# A 60-Day Green Tea Extract Supplementation Counteracts the Dysfunction of Adipose Tissue in Overweight Post-Menopausal and Class I Obese Women

**DOI:** 10.3390/nu14245209

**Published:** 2022-12-07

**Authors:** Mariangela Rondanelli, Clara Gasparri, Simone Perna, Giovanna Petrangolini, Pietro Allegrini, Teresa Fazia, Luisa Bernardinelli, Alessandro Cavioni, Francesca Mansueto, Letizia Oberto, Zaira Patelli, Alice Tartara, Antonella Riva

**Affiliations:** 1IRCCS Mondino Foundation, 27100 Pavia, Italy; 2Unit of Human and Clinical Nutrition, Department of Public Health, Experimental and Forensic Medicine, University of Pavia, 27100 Pavia, Italy; 3Endocrinology and Nutrition Unit, Azienda di Servizi alla Persona ‘‘Istituto Santa Margherita’’, University of Pavia, 27100 Pavia, Italy; 4Department of Biology, College of Science, University of Bahrain, Sakhir Campus, Zallaq P.O. Box 32038, Bahrain; 5Research and Development Department, Indena SpA, 20139 Milan, Italy; 6Department of Brain and Behavioral Science, University of Pavia, 27100 Pavia, Italy

**Keywords:** green tea extract, dietary supplement, obesity, menopausal women, phytosome

## Abstract

Menopause is characterized by weight gain and increased visceral fat, which acts as an endocrine organ secreting proinflammatory adipocytokines, with consequent increased risk of metabolic disorders. The aim of this double-blind, placebo-controlled randomized trial was to evaluate the effects of a 60-day dietary supplementation using *Camellia sinensis* leaf extract on adipose tissue dysfunction in overweight or class I obese post-menopausal, sedentary women. Primary endpoints were the respiratory quotient (RQ), the percentage of carbohydrates (%CHO), the percentage of fat oxidation (%FAT), and the resting energy expenditure (REE) measured by indirect calorimetry. Secondary endpoints included body composition, by dual x-ray absorptiometry (DXA), glucose profile, lipid profile, inflammatory state, liver and kidney function, hormonal status regarding satiety, and status of catecholamines. Twenty-eight women were randomized into two groups: 14 (BMI 31.1 ± 3.5) were supplemented and 14 (BMI 31.9 ± 2.2) received placebo. In regards to the between-group differences over time (β), a statistically significant difference between the supplemented and placebo group was observed for: RQ (β = −0.04, *p* = 0.009), % fat oxidation (β = 11.04, *p* = 0.0006), insulin (β = −1.74, *p* = 0.009), HOMA (β = −0.31, *p* = 0.02), waist circumference (β = −1.07, *p* = 0.007), REE (β = 83.21, *p* = 0.009), and CRP (β = −0.14, *p* = 0.02). These results demonstrate that a 60-day green tea extract supplementation counteracts the dysfunction of adipose tissue in overweight post-menopausal and class I obese women.

## 1. Introduction

The visceral fat, acting as an endocrine organ secreting proinflammatory adipocytokines, increased at the onset of menopause, together with body weight; therefore a consequent greater risk of metabolic disorders such as cardiovascular diseases and type 2 diabetes was present in menopausal women [1].

Body fat redistribution changed in post-menopausal women with a marked increase in abdominal fat [2] that may augment the development of hepatic insulin resistance [3].

Levels of various hormones change during menopause in addition to sex hormones; adiponectin levels are reduced and this reduction is one of the causes of increased adipose tissue [4]. The increase in adipose tissue may be associated with the onset of both hepatic and peripheral insulin resistance [5]. Therefore, the adiponectin decrease may worsen metabolic dysregulation, and exacerbate lipid-induced insulin resistance in the liver and skeletal muscle [6,7].

This dysfunction of adipose tissue can be counteracted through the support of dietary supplements in combination with a low-calorie diet [8]. Among dietary supplements, tea leaf extracts are spreading in use for their metabolic properties.

Green tea prepared from Camellia sinensis [9], contains polyphenols up to 35% of the dry weight, mainly flavanols, flavones, and flavan-3-ols [9]. The well-known compounds of the last group are the catechins, with epigallocatechin-3-gallate (EGCG) being the most abundant and the most bioactive component of green tea, representing 50–80% of the total catechins content [10,11]. Caffeine, thearubigins, theanine, theaflavins, quercetin, and other phenolics such as gallic acid and chlorogenic acid, together with other minor catechins (epicatechin-3-gallate 222(ECG), epigallocatechin (EGC), epicatechin and catechin) were also contained in green tea [9,11,12]. 

Green tea catechins (GTCs) may affect multiple aspects of energy and hormone balance that lead to body weight and fat loss, so they are considered the most responsible for beneficial effects on obesity [13]. A meta-analysis on this topic supports the findings that EGCG has an effect on metabolic parameters; specifically, the results revealed that EGCG intake moderately accelerates resting energy expenditure (REE) and reduces the respiratory quotient (RQ), even at a low dose of 300 mg/day [14]. These observations have been confirmed by a recent systematic review that considered both chronic and acute intake of GTC-based supplements [15]. The decrease of RQ occurred even at a low dose of GTC (100–300 mg/day); this finding suggests that EGCG has the potential to increase fat oxidation and may thereby contribute to the anti-obesity effects of green tea [15].

There are several proposed mechanisms through which GTC may influence body weight and body composition. The predominating hypothesis is that GTC influences sympathetic nervous system (SNS) activity, increasing energy expenditure and promoting the oxidation of fat. 

Other potential mechanisms include decreased nutrient absorption, modifications in appetite and up-regulation of enzymes involved in hepatic fat oxidation [9].

In addition, a recent meta-analysis suggests that consumption of green tea supplementation in obese subjects with metabolic syndrome has beneficial effects on the improvement of lipid and glucose metabolism, as well as in the facilitation of weight loss [16].

Given this background, the aim of this randomized clinical trial was to evaluate the effects of a 60-day dietary supplementation with *Camellia sinensis* leaf extract formulated in phospholipids on indicators of adipose tissue dysfunction in overweight or class I obese post-menopausal sedentary women. We considered as a primary endpoint RQ, carbohydrates oxidation, FAT oxidation, and REE measured by indirect calorimetry. Secondary endpoints included body composition (Fat Free Mass (FFM), Fat Mass (FM), and visceral adipose tissue (VAT)) measured by dual-energy X-ray absorptiometry (DXA), glucose profile (glycaemia, insulinemia and Homeostasis Model Assessment (HOMA) index), lipid profile (total cholesterol (TC), high-density lipoprotein-cholesterol (HDL-C), low-density lipoprotein-cholesterol (LDL-C) and triglycerides (TG)), anti-/proinflammatory state (adiponectin, C reactive protein (CRP)), liver and kidney function (alanine aminotransferase (ALT), aspartate aminotransferase (AST), Gamma-Glutamyl Transferase (G-GT), creatinine), hormonal status regarding satiety (leptin), and status of catecholamines (adrenaline and noradrenaline). 

## 2. Methods

### 2.1. Population

A randomized, double-blind, placebo-controlled trial was conducted in overweight or obese post-menopausal women. The subjects were recruited from the Dietetic and Metabolic Unit of the “Santa Margherita” Institute, University of Pavia, Italy. 

The subjects, without past cardiovascular disease (CVD) and free of overt liver, renal and thyroid diseases, were not taking any medication likely to affect glucose or lipid metabolism (oral hypoglycemic agents and statins). The subjects who smoked or drank more than two standard alcoholic beverages/day (20 g of alcohol/day) were excluded from the study. Physical activity was recorded. Sedentary subjects were admitted into the study. The experimental protocol was approved by the Ethics Committee of the University of Pavia (ethical code Number: 0905/14122018) and was registered at ClinicalTrials.gov under registration number: NCT05031702. All the volunteers gave their written informed consent to participate.

### 2.2. Dietary Supplement

The dietary supplementation was associated with 2 daily oral doses (one before lunch and one before dinner) of 150 mg of Greenselect^®^ Phytosome^®^ (GSP, Indena S.p.A, Milan, Italy) standardized to contain ≥19.0% ≤25.0% catechins by HPLC, ≥13.0% of epigallocatechin-3-O-gallate (EGCG) by HPLC, ≤0.1% of caffeine by HPLC. The supplementation period was 60 days.

Tablets containing verum and placebo were provided by Indena SpA (Milan, Italy). Placebo tablets with no active ingredient were used and were identical to verum ones, in terms of size, shape, color, odor and taste. Verum and placebo film-coated tablets had similar compositions with regards to inactive food-grade components. Before release, the film-coated tablets were tested for appearance, average mass, uniformity of mass, HPLC-content of GSP active compounds, disintegration time, and microbiological quality. All procedures were performed according to Food Supplement Regulations in the European Union. 

The number of tablets actually taken by each subject, divided by the number of tablets that should have been taken during the study was recorded as compliance to the supplementation regimen. 

Identical products for each supplemented group were assigned to a subject number according to a coded (AB) block randomization table prepared by an independent statistician. Investigators were blinded to the randomization table, the code assignments, and the procedure. All the subjects followed a similar low energy diet. Regarding blinding, the active supplementation and placebo were given in identical containers devoid of any labelling by the principal investigator, who was not involved in any of the assessments.

### 2.3. Adverse Events

Adverse events were collected by spontaneous reporting by subjects, as well as open-ended inquiries by members of the research staff. Moreover, routine blood biochemistry parameters (i.e., creatinine, liver function) were evaluated at the start and at the end of supplementation.

### 2.4. Blood Parameters

The glycemic and lipidic parameters were assessed at baseline (t_0_), after 30 days (t_1_), and after 60 days at the end of supplementation (t_2_). Blood samples were obtained through an indwelling catheter inserted in an antecubital vein. The following parameters were evaluated: the glucose profile (glycaemia, insulinemia and HOMA index), the lipid profile (total cholesterol, HDL-C, LDL-C and TG), the inflammatory state (adiponectin, CRP), the liver and kidney function (AST, ALT, GGT, creatinine); the hormonal status regarding satiety (leptin) and the status of circulating catecholamines (adrenaline and noradrenaline) were carried out at t_0_. The second venous sampling was fixed after 30 days and consisted of the evaluation of the glucose profile (glycaemia, insulinemia and HOMA index), the anti-/proinflammatory state (adiponectin, CRP), the hormonal state concerning satiety (leptin), and the state of circulating catecholamines (adrenaline and noradrenaline). The third venous sampling was fixed after 60 days and consisted of the same evaluations as the first sampling.

Participants were asked to collect 24 h-urine, in order to evaluate azoturia at baseline (t_0_), after 30 days (t_1_), and after 60 days at the end of supplementation (t_2_).

Serum levels of leptin and adiponectin were measured using enzyme-linked immunosorbent assay DuoSet Kits (R&D Systems, Minneapolis, MN, USA) according to the manufacturer’s instructions.

Fasting blood glucose (FBG), TC, LDL-C, HDL-C and TG levels were measured by an automatic biochemical analyzer (Hitachi 747, Tokyo, Japan). The serum insulin was evaluated by a double antibody RIA (Kabi Pharmacia Diagnostics AB, Uppsala, Sweden) and expressed as pmol/L. The intra- and inter-assay coefficients of variation were below 6% and the low detection limit was 10.7 pmol/L. To determine insulin resistance, subjects were instructed to fast for 12 h before obtaining the blood sample. Furthermore, the subjects refrained from any form of physical exercise for 48 h before blood sampling. Insulin resistance was evaluated using the HOMA index [17].

### 2.5. Anthropometric Mesurements and Dietary Counselling

Anthropometric measurements at the start of the study at baseline (t_0_), after 30 days (t_1_) and after 60 days at the end of supplementation (t_2_) were made to assess nutritional status. Body weight and height were measured following a standardized technique [18] and the BMI was calculated (kg/m^2^). 

Subjects consumed a hypocaloric diet useful to maintain a prudent balance of macronutrients: 25–30% of energy from fat (cholesterol < 200 mg), 55–60% of energy from carbohydrates (10% from simple carbohydrates), and 15–20% of energy from protein [19]. A registered dietician performed initial dietary counselling. A visual-analogue scale to evaluate the sense of hunger/satiety was administered for the entire duration of the study.

### 2.6. Body Composition

Body composition (FFM, FM, and gynoid and android fat distribution) was measured by DXA with the use of a Lunar Prodigy DXA (GE Medical Systems). The in vivo CVs were 0.89% and 0.48% for whole body fat (FM) and FFM, respectively.

VAT volume was estimated using a constant correction factor (0.94 g/cm^3^). The software automatically places a quadrilateral box, which represents the android region, outlined by the iliac crest and with a superior height equivalent to 20% of the distance from the top of the iliac crest to the base of the skull [20].

### 2.7. Assessment of Resting Energy Expenditure (REE)

Respiratory exchange measurements using indirect calorimetry (Q-NRG, Cosmed, Rome, Italy) were used to estimate REE, adhering to the recommended measurement conditions [21].

REE was calculated from O_2_ and CO_2_ volumes—as well as from urine excretion nitrogen values—using the Weir formula and expressed as kcal/day to obtain post-prandial RQ and substrate oxidation, and continuous gas exchange was determined [22]. CHO and FAT oxidation were also evaluated by indirect calorimetry.

### 2.8. Primary and Secondary Endpoints

We considered as primary endpoints RQ, CHO and FAT oxidation, and REE measured by indirect calorimetry. 

Secondary endpoints were body composition (FFM, FM, and VAT), measured by DXA, glucose profile (glycaemia, insulinemia and HOMA index), lipid profile (TC, HDL-C, LDL-C and TG), anti/proinflammatory state (adiponectin, CRP), liver and kidney function (ALT, AST, G-GT, creatinine), hormonal status regarding satiety (leptin), and status of catecholamines (adrenaline and noradrenaline). 

### 2.9. Statistical Analysis

Differences between groups at baseline were investigated in each continuous variable by using t-test for independent data in normally distributed variables or Wilcoxon test in non-normally distributed ones, and by using chi-squared for categorial variables. 

Linear mixed models (LMM) [23] were used to evaluate the between- and within- group differences for primary and secondary endpoints. To evaluate statistically significant changes over time between the two groups (GSP and placebo) in the LMM, time group, and the interaction time*group were specified as fixed effect, while a random intercept for each subject, in the form of 1|subject, was added to account for the intrasubject correlation produced by the different measurements carried out on the same patients. The coefficient of the interaction between time and group (β Time*group) was estimated for measuring the difference in slopes between the two groups indicating how much more the GSP group is improving over time with respect to the investigated endpoints, compared to the placebo group over the same period. With regards to the within-group differences, the coefficient of time (β Time) in each group was estimated and indicates the change over time in the considered group. All the models were adjusted for age and BMI. *p*-values < 0.05 on the 2-sided test were considered as statistically significant. Normality was graphically assessed with the Shapiro-Wilk test. In case of non-normality of LMM residuals, empirical bootstrap with 1000 bootstrapped replicates was applied to estimate non-parametric 95% C.I.s and *p*-values based on distribution’s quantiles [24]. Benjamini-Hochberg correction, fixing the false discovery rate (FDR) at alpha <0.05, was used to account for multiple comparisons [25].

Pearson’s pairwise correlations (*r*) and their corresponding *p*-values, were also calculated to investigate the linear relationship between selected variables both at pre- and post-supplementation.

Descriptive statistics are reported as Mean ± Standard Deviation (SD). All the analysis was performed on R 3.5.1 software using the nlme and stats packages [26].

## 3. Results

A total of 28 women, with mean (±SD) age of (58.75 ± 6.68) were recruited and randomly assigned to the placebo group (*n* = 14) or supplemented group (*n* = 14). Baseline (t_0_) characteristics of participants are shown in Table 1. No statistically significant differences were observed between the two groups at baseline parameters.

Table 2 reports the descriptive statistics in terms of mean ± SD for each endpoint measured in the two groups at t_0_, after 30 days (t_1_) and after 60 days (t_2_). 

Results of the within-group and between-group differences, adjusted for age and BMI are reported respectively in Table 3 and Table 4. In regards to the within-group differences (Table 3), after multiple testing correction, the results showed in the GSP group a statistically significant decrease in RQ (β = −0.03, 95%C.I. [−0.04;−0.02], *p* < 0.0001), insulin (β = −1.29, 95%C.I. [−2.01;−0.49], *p* = 0.007), HOMA (β = −0.25, 95%C.I. [−0.40;−0.09], *p* = 0.01), waist circumference (β = −1.66, 95%C.I. [−2.20;−1.13], *p* < 0.0001), VAT (β = −85.25, 95%C.I. [−126.39;−44.11], *p* = 0.0008), fat mass (β = −1025.64, 95%C.I. [−1331.97;−719.31], *p* < 0.0001), %CHO (β = −13.14, 95%C.I. [−17.33;−8.96], *p* < 0.0001) and CRP (β = −0.15, 95%C.I. [−0.23;−0.06], *p* = 0.007), and a statistically significant increase in adiponectin (β = 1.54, 95%C.I. [0.42;2.49], *p* = 0.02), noradrenalin (β = 59.07, 95%C.I. [28.78;89.36], *p* = 0.001), MB (β = 67.75, 95%C.I. [27.67;107.83], *p* = 0.006), %LIP (β = 13.14, 95%C.I. [8.96;17.33], *p* < 0.0001), and adiponectin/leptin ratio (β = 0.06, 95%C.I. [0.04;0.09], *p* = 0.0001) after 60 days. As for the placebo group, after multiple testing correction, results showed a statistically significant decrease in waist circumference (β = −0.58, 95%C.I. [−0.90; −0.28], *p* = 0.02) and a statistically significant increase in adiponectin/leptin ratio (β = 0.05, 95%C.I. [0.02;0.08], *p* = 0.04) after 60 days. In regards to the between-group differences (Table 4), a statistically significant interaction between time and group, meaning that the change over time is different for each group, was observed for insulin (β = −1.74, 95%C.I. [−2.71;−0.65], *p* = 0.009), HOMA (β = −0.31, 95%C.I. [−0.51;−0.09], *p* = 0.02), CRP (β = −0.14, 95%C.I. [−0.24;−0.05], *p* = 0.02), waist circumference (β = −1.07, 95%C.I. [−1.68;−0.47], *p* = 0.007), REE (β = 83.21, 95%C.I. [31.33;135.10], *p* = 0.009), RQ (β = −0.04, 95%C.I. [−0.07;−0.02], *p* = 0.009), %CHO (β = −11.04, 95%C.I. [−16.03;−6.04], *p* = 0.0006), and %LIP (β = 11.04, 95%C.I. [6.04;16.03], *p* = 0.0006).

## 4. Discussion

This is the first study in the literature that evaluated the use of substrates, in addition to REE and RQ, measured by indirect calorimetry, after Camellia sinensis leaf extract formulated in phospholipids as a dietary supplementation in overweight or class I obese post-menopausal sedentary women. The main finding of this double-blind placebo-controlled clinical study was a statistically significant difference between supplemented and placebo group for RQ, %CHO, and %LIP.

Previous studies with administration of green tea extract have already reported significant improvements in fat oxidation rates, but the subjects were men and untrained, overweight individuals at rest [27,28,29]. 

The results of this study are interesting because the stimulation of lipid catabolism represents one way to maintain the correct body weight [30].

It is possible that the stimulation of lipid catabolism could also be due to the significant increase in norepinephrine demonstrated in the GSP group, but not in the placebo group.

Catecholamines are known to increase the mobilization of stored triglycerides in adipocytes to release fatty acids (FAs) for other tissues. Sympathetic activation can increase both lipolysis and FA oxidation in adipocytes, revealing a new regulatory axis in metabolism. In animal models, it has been demonstrated that catecholamines suppress fatty acid re-esterification and increase oxidation in white adipocytes via STAT3 [31].

An exhaustive review by Rains [9] reported that polyphenols of green tea can influence SNS activity, by increasing energy expenditure and promoting the oxidation of fat. Modifications in appetite, reduced nutrient absorption, and up-regulation of enzymes involved in hepatic fat oxidation are other potential mechanisms [9].

The significant reduction of RQ, found in the supplemented group only, is in agreement with a previous review [15], and is an interesting result because RQ, as a reflection of carbohydrate and fat oxidation, has been proposed to be a metabolic index predicting subsequent weight gain [32,33,34,35]. Obesity is believed to be a proinflammatory disease on the basis of extensive research over the past two decades [36]. 

Our results regarding inflammatory biomarkers, such as CRP and VAT which are significantly reduced in the supplemented group only, were supported by a recent review by Ntamo [37], who reported that preclinical evidence indicates that Epigallocatechin gallate, one of the most abundant and powerful flavonoids contained in green tea, has strong anti-inflammatory properties [37].

The development of insulin resistance is one of the consequences of this state of inflammation in obesity [38]. All the patients who participated in the study showed insulin resistance, as demonstrated by the HOMA index values at baseline. In regard to the statistically significant reduction of the HOMA index found in the GSP group only, this could be interpreted in light of a recent paper that investigated, in vivo, the influence of epigallocatechin-3-gallate (EGCG) and its autoxidation products on insulin sensitivity via renin-angiotensin system (RAS) regulation.

Our results are in line with other clinical studies, although conducted in subjects with different characteristics, such as patients with Nonalcoholic Fatty Liver Diseases (NAFLD) [39], and in animal models [40].

Considering body composition evaluated by DXA, the treated group had a statistically significant decrease in VAT and fat mass, while the FFM was not changed. Therefore, 60 days of green tea supplementation may cause positive effects on body composition. In fact, it has been found that green tea induced a redistribution of adipose tissue, by reducing visceral fat mass and fat mass, in the absence of a fat-free mass reduction, as also already shown by Nagao’s study [41].

Moreover, numerous studies indicate that green tea dietary supplementation has a number of potential beneficial effects for the reduction of body weight, even if body composition is not assessed [42,43,44].

Finally, another interesting result of this study is the statistically significant increase in adiponectin and the adiponectin/leptin ratio of the treated group. These results are in line with those obtained from a previous randomized controlled trial (RCT) study by Chen et al. [43]; the study performed in a group of overweight women for 12 weeks showed that adiponectin levels were significantly elevated in the group treated with a high-dose green tea extract (EGCG at a daily dosage of 856.8 mg) and not in the placebo group [43].

Adiponectin is an adipocyte-derived hormone believed to sensitize insulin, improving the energy metabolism of tissues [45], protecting against excessive hepatic lipid accumulation and exerting anti-inflammatory effects [46]. Green tea catechins have been shown to upregulate adiponectin expression in mouse pre-adipocyte cells [47]. 

In other studies, green tea supplementation does not appear to affect obesity hormones, leptin, and adiponectin. This inconsistency can be explained by the fact that in these studies a sufficient loss of fat mass was not found [48]. Furthermore, these studies were conducted in populations not enrolled in the study reported here, such as obese subjects with metabolic syndrome [49], type 2 diabetics [50], young (25- to 35-year-old), obese women after weight loss [51], or in obese adult subjects (males and females of childbearing age) on diets with extremely low daily food energy consumption [52].

The strength of this study lies in the collection and analysis of a variety of factors that can determine weight loss and changes in body composition after taking a green tea extract. It is worth noting that the supplement used in this study is a lecithin delivery formulated of a highly standardized caffeine-free green tea extract that allows an ameliorated oral bio-absorption of epigallocatechin 3-O-gallate (EGCG) [53]. Previous clinical studies showed interesting weight management properties [54,55] substantiated now with a strong mechanism of action thanks to our new data. The main limitation of this study is that it was not possible to calculate protein oxidation, as we did not measure urinary nitrogen excretion. Moreover, another limitation of this study is that a specific population was enrolled (overweight and class I obese post-menopausal sedentary women), and the results cannot be extended to other populations. Finally, another limitation is the number of subjects studied: although it is a double-blind randomized study against placebo, the number of subjects is relatively limited, so more extensive research is required.

## 5. Conclusions

The results of this study demonstrate beneficial changes in the lipolysis pathway, especially in weight and body fat reduction, in subjects supplemented with a green tea extract formulated in phospholipids with respect to the placebo group. The indicated 60-day green tea extract supplementation counteracts the dysfunction of adipose tissue in post-menopausal overweight and class I obese women.

## Figures and Tables

**Table 1 nutrients-14-05209-t001:** Mean (SD) at baseline (t_0_) for age and BMI, and *p*-value of the difference between the two groups.

	Placebo Group (*n* = 14) Mean (SD)	GSP Group (*n* = 14) Mean (SD)	*p*-Value
Age (y)	56.92 (5.70)	60.57 (7.28)	0.15
BMI (Kg/m^2^)	31.10 (3.54)	31.99 (2.23)	0.84

**Table 2 nutrients-14-05209-t002:** Descriptive statistics for each endpoint measured in the two groups at baseline (t_0_), after 30 days (t_1_) and after 60 days (t_2_).

	Placebo Group (*n* = 14) Mean (SD)			GSP Group (*n* = 14) Mean (SD)		
	t_0_	t_1_	t_2_	t_0_	t_1_	t_2_
Primary endpoints						
RQ	0.79 (0.09)	0.79 (0.09)	0.81 (0.05)	0.77 (0.06)	0.72 (0.05)	0.71 (0.05)
%CHO oxidation	48.21 (16.57)	49.21 (16.22)	44.00 (18.83)	47.50 (18.28)	21.14 (12.50)	21.21 (12.66)
%LIP oxidation	51.79 (16.57)	50.79 (16.22)	56.00 (18.83)	52.50 (18.28)	78.86 (12.50)	78.79 (12.66)
REE (kcal/day)	1396.57 (238.37)	1390.71 (201.04)	1365.64 (230.96)	1401.71 (202.88)	1500.64 (229.12)	1537.21 (190.76)
Secondary endpoints						
Body composition						
Fat Free Mass (g)	43,033.43 (4005.97)	43,505.93 (3728.92)	43,013.21 (3595.07)	41,353.36 (3587.18)	41,377.00 (3736.21)	40,872.43 (3333.17)
Fat Mass (g)	38,630.71 (9224.67)	38,019.00 (10,345.88)	37,543.36 (10,608.83)	37,882.86 (6013.62)	36,472.36 (5819.56)	35,831.57 (5488.24)
VAT (g)	1395.71 (636.83)	1370.71 (665.94)	1298.56 (832.95)	1383.21 (429.11)	1235.64 (489.22)	1212.71 (441.54)
Waist Circumference (cm)	101.25 (9.83)	100.32 (10.14)	100.07 (10.17)	105.86 (7.35)	104.57 (6.49)	102.54 (6.47)
Glucose and lipid profile						
Glycemia (mg/dL)	90.36 (15.35)	89.14 (12.91)	87.43 (12.15)	87.71 (8.94)	90.36 (8.85)	88.71 (8.44)
Insulin (μIU/mL)	11.04 (4.24)	11.30 (4.21)	11.94 (5.03)	11.88 (5.09)	9.97 (4.01)	9.30 (3.60)
HOMA (pt)	2.50 (1.10)	2.54 (1.15)	2.61 (1.20)	2.54 (1.07)	2.21 (0.86)	2.04 (0.81)
Total Cholesterol (mg/dL)	194.86 (37.08)	-	199.21 (36.36)	199.86 (30.86)	-	199.07 (28.37)
HDL (mg/dL)	61.50 (18.18)	-	61.00 (16.85)	66.86 (12.53)	-	65.14 (10.83)
LDL (mg/dL)	116.00 (30.44)	-	115.71 (30.90)	115.93 (22.67)	-	120.36 (21.67)
VLDL (mg/dL)	22.74 (6.26)	-	25.50 (13.13)	23.34 (8.27)	-	24.06 (7.39)
Triglycerides (mg/dL)	113.71 (31.29)	-	127.50 (65.64)	116.71 (41.35)	-	120.29 (36.93)
Anti-/proinflammatory state						
Adiponectin (μg/mL)	8.12 (4.24)	8.61 (4.17)	9.56 (5.15)	7.50 (2.95)	9.76 (4.93)	10.59 (6.16)
Adiponectin/leptin ratio	0.23 (0.15)	0.27 (0.18)	0.32 (0.26)	0.18 (0.10)	0.26 (0.15)	0.30 (0.16)
CRP (mg/L)	0.29 (0.22)	-	0.29 (0.22)	0.30 (0.22)	-	0.16 (0.13)
Liver and kidney function						
AST (IU/l)	19.36 (6.21)	-	17.14 (4.05)	17.00 (5.13)	-	18.21 (4.63)
ALT (IU/l)	20.79 (8.92)	-	18.21 (8.76)	16.50 (6.56)	-	19.21 (5.06)
G-GT (U/l)	23.07 (17.41)	-	24.50 (25.97)	15.79 (7.08)	-	16.00 (6.96)
Creatinine (mg/dl)	0.78 (0.07)	-	0.82 (0.09)	0.69 (0.13)	-	0.71 (0.15)
Azoturia (g/24 h)	57.04 (16.87)	47.17 (18.56)	50.15 (17.22)	51.82 (12.69)	55.58 (14.84)	52.88 (22.76)
Hormonal profile and status of catecholamines						
Leptin (ng/mL)	40.45 (16.97)	37.51 (17.38)	37.30 (19.61)	48.61 (17.71)	43.26 (15.92)	39.78 (15.35)
Adrenaline (pg/mL)	26.93 (11.49)	27.00 (19.91)	26.57 (19.25)	31.07 (17.18)	31.29 (14.37)	36.93 (20.62)
Noradrenaline (pg/mL)	485.43 (185.13)	393.21 (145.13)	486.71 (128.97)	509.50 (186.25)	571.50 (245.84)	627.64 (279.69)

- = no data was collected at this time point. Abbreviations: HOMA: Homeostasis Model Assessment; VAT: visceral adipose tissue; LDL: Low-density lipoprotein; HDL: High-density lipoprotein; ALT: alanine aminotransferase; AST: aspartate aminotransferase; G-GT: Gamma-Glutamyl Transferase (G-GT); CRP: C reactive protein.

**Table 3 nutrients-14-05209-t003:** Within-group pre-post supplementation difference for primary and secondary endpoints. Estimate of time (β), 95% confidence interval (CI) and the adjusted *p*-value of the null hypothesis of no effect are reported for the two groups.

	Placebo Group		GSP Group	
	Time β [95%CI]	*p*-Value Adjusted	Time β [95%CI]	*p*-Value Adjusted
Primary endpoints				
RQ	0.01 [−0.01;0.04]	0.47	−0.03 [−0.04;−0.02]	<0.0001
%CHO	−2.11 [−5.42;1.67]	0.47	−13.14 [−17.33;−8.96]	<0.0001
%LIP	−2.11 [−1.67;5.42]	0.47	13.14 [8.96;17.33]	<0.0001
REE (kcal/day)	−15.46 [−50.29;19.37]	0.56	67.75 [27.67;107.83]	0.006
Secondary endpoints				
Body composition				
Fat Free Mass (g)	−10.11 [−335.77;315.55]	0.99	−240.46 [−587.19;106.26]	0.24
Fat Mass (g)	−543.68 [−997.30;−90.06]	0.18	−1025.64 [−1331.97;−719.31]	<0.0001
VAT (g)	−48.58 [−163.47;102.00]	0.63	−85.25 [−126.39;−44.11]	0.0008
Waist Circumference (cm)	−0.58 [−0.90;−0.28]	0.02	−1.66 [−2.20;−1.13]	<0.0001
Glucose and lipid profile				
Glycemia (mg/dL)	−1.46 [−3.65;0.72]	0.47	0.50 [−1.36;2.36]	0.72
Insulin (μIU/mL)	0.45 [−0.29;1.08]	0.47	−1.29 [−2.01;−0.49]	0.007
HOMA (pt)	0.06 [−0.06;0.17]	0.49	−0.25 [−0.40;−0.09]	0.01
Total Cholesterol (mmol/L)	4.36 [−8.37;17.09]	0.63	−0.79 [−7.09;5.52]	0.79
HDL (mg/dL)	−0.50 [−5.52;4.52]	0.99	−1.71 [−4.15;0.72]	0.23
LDL (mg/dL)	−0.29 [−10.97;10.40]	0.99	4.43 [−4.52;13.38]	0.40
VLDL (mg/dL)	2.76 [−2.52;8.03]	0.47	0.71 [−3.82;5.25]	0.78
Triglycerides (mg/dL)	13.79 [−12.58;10.15]	0.47	3.57 [−24.21;30.07]	0.78
Anti-/proinflammatory state				
Adiponectin (μg/mL)	0.72 [−0.56;1.68]	0.47	1.54 [0.42;2.49]	0.02
Adiponectin/leptin ratio	0.05 [0.02;0.08]	0.04	0.06 [0.04;0.09]	0.0001
CRP (mg/L)	−0.004 [−0.05;0.04]	0.99	−0.15 [−0.23;−0.06]	0.007
Liver and kidney function				
AST (IU/l)	−2.21 [−5.32;0.89]	0.47	1.21 [−0.42;2.85]	0.21
ALT (IU/l)	−2.57 [−5.78;0.64]	0.47	2.71 [−0.11;5.54]	0.11
G-GT (U/l)	1.43 [−6.65;9.51]	0.91	0.21 [−0.90;1.33]	0.78
Creatinine (mg/dl)	0.04 [−0.02;0.09]	0.47	0.03 [−0.006;0.06]	0.17
Azoturia (g/24 h)	−3.45 [−8.39;1.50]	0.47	0.53 [−2.93;4.00]	0.78
Hormonal profile and status of catecholamines				
Leptin (ng/mL)	−1.55 [−3.86;0.71]	0.47	−4.42 [−7.99;0.06]	0.10
Adrenaline (pg/mL)	−0.18 [−5.88;5.03]	0.99	2.93 [−3.62;10.08]	0.55
Noradrenaline (pg/mL)	0.64 [−61.46;62.75]	0.99	59.07 [28.78;89.36]	0.001

**Table 4 nutrients-14-05209-t004:** Between-group pre-post supplementation difference for primary and secondary endpoint. Estimate of time*treatment (β), 95% confidence interval (CI) and the adjusted *p*-value of the null hypothesis of no effect are reported.

	Time*Group β [95%CI]	*p*-Value Adjusted
Primary endpoints		
RQ	−0.04 [−0.07;−0.02]	0.009
REE (kcal/day)	83.21 [31.33;135.10]	0.009
%CHO	−11.04 [−16.03;−6.04]	0.0006
%LIP	11.04 [6.04;16.03]	0.0006
Secondary endpoints		
Body composition		
Fat Free Mass (g)	−230.36 [−695.15;234.44]	0.51
Fat Mass (g)	−481.96 [−1016.80;52.87]	0.22
VAT (g)	−36.67 [−194.60;86.15]	0.73
Waist Circumference (cm)	−1.07 [−1.68;−0.47]	0.007
Glucose and lipid profile		
Glycemia (mg/dL)	1.96 [−0.84;4.77]	0.35
Insulin (μIU/mL)	−1.74 [−2.71;−0.65]	0.009
HOMA (pt)	−0.31 [−0.51;−0.09]	0.02
Total Cholesterol (mmol/L)	−5.14 [−21.58;13.19]	0.66
HDL (mg/dL)	−1.21 [−7.28;6.21]	0.73
LDL (mg/dL)	4.71 [−8.55;17.97]	0.66
VLDL (mg/dL)	−2.04 [−10.11;5.90]	0.66
Triglycerides (mg/dL)	−10.21 [−50.57;29.51]	0.66
Anti-/proinflammatory state		
Adiponectin (μg/mL)	0.83 [−0.72;2.46]	0.51
Adiponectin/leptin ratio	0.01 [−0.02;0.05]	0.66
CRP (mg/L)	−0.14 [−0.24;−0.05]	0.02
Liver and kidney function		
AST (IU/l)	3.43 [−0.67;7.55]	0.23
ALT (IU/l)	5.29 [0.03;10.34]	0.15
G-GT (U/l)	−1.21 [−9.23;9.67]	0.73
Creatinine (mg/dl)	−0.01 [−0.07;0.05]	0.73
Azoturia (g/24 h)	3.98 [−1.92;9.88]	0.35
Hormonal profile and status of catecholamines		
Leptin (ng/mL)	−2.84 [−7.38;2.37]	0.47
Adrenaline (pg/mL)	3.11 [−5.27;11.75]	0.66
Noradrenaline (pg/mL)	58.43 [−10.23;128.05]	0.22

## Data Availability

Not applicable.

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
