# Peer review of "A 60-Day Green Tea Extract Supplementation Counteracts the Dysfunction of Adipose Tissue in Overweight Post-Menopausal and Class I Obese Women"

_nutrients, 2022, doi:10.3390/nu14245209_

Round 1
Reviewer 1 Report
This randomized clinical trial was aimed to evaluate the effects of a 60-day dietary supplementation with Camellia sinensis leaf extract formulated in phospholipids on indicators of adipose tissue dysfunction in overweight or class I obese post-menopausal sedentary women. Twenty-eight women were randomized into two groups: supplemented group (n=14) and placebo group (n=14). The results showed that In regards to the between-group differences over time (β), a statistically significant difference was observed for the Primary endpoint: the respiratory quotient (RQ), the resting energy expenditure (REE), the percentage of carbohydrates oxidation (%CHO), and the percentage of fat oxidation (%LIP) as well as for the Secondary endpoints: waist circumference, insulin, HOMA, and CRP. There are some concerns as listed in the following:
(1) It is better to match the sequence of the statement, e.g. anti/proinflammatory state (CRP, adiponectin) -> anti/proinflammatory state (adiponectin, CRP); liver and kidney function (creatinine, ALT, AST, G-GT)-> liver and kidney function (ALT, AST, G-GT, creatinine) [check all]
(2) It is unclear why a significant change of CRP level between the supplemented group and placebo group (p=0.02, Table 4) was not mentioned in the Results and the Abstract.
(3) Keep consistent writing format for the References, including: [check all] e.g.
*Author: R4: Rolland YM, Perry HM 3rd, Patrick P, Banks WA, M.J.; R52: Ryu OH, L.J.L.K.K.H.S.J.K.S.K.N.B.S.C.D.C.K.
*Title: R6: Decreased Plasma Adiponectin Concentrations Are.. vs. Decreased plasma adiponectin concentrations are..
*Journal name: R46: J Am Coll Nutr. vs. J. Am. Coll. Nutr.
*Page number: R4: 1630–6. vs 1630–1636
(4) Typos ad others:
*L33-36: no CRP (p=0.02)?
L98: (C reactive protein (CRP), adiponectin) sequence?
L98-99: liver and kidney function (creatinine, alanine ami- 98 notransferase (ALT), aspartate aminotransferase (AST), Gamma-Glutamyl Transferase (G-GT))
L150: (creatinine, AST, ALT, GGT),
L155: (CRP, adiponectin)
L179: protein. [20].
L186: (0.94 g/cm3).
L196: oxidation, continuous gas exchange was determined
L203-204: anti/proinflammatory state (CRP, adiponectin), liver and kidney function (creatinine, ALT, AST, G-GT)
L211-212: we fitted
*L253: and PCR -> and CRP
*L263-267: CRP? (p=0.02)
*L264: MB -> REE (kcal/day)
*L268: Secondary enpoints
*L275: Secondary enpoints
*L415: Clin. Nutr. 2019. [Clin Nutr. 2020 Apr;39(4):1049-1058. doi: 10.1016/j.clnu.2019.05.019. Epub 2019 May 24.]
*L420: U Expert Consultation; 1985;
*L422: Nutr. Diabetes 2017, 7 [Nutr Diabetes . 2017 Jan 9;7(1):e238. doi: 10.1038/nutd.2016.38.]
*L431: computing; 2017;
*L462: Antioxidants 2021, 10. [Antioxidants (Basel). 2021 Jul 5;10(7):1076. doi: 10.3390/antiox10071076.
*L496: Evidence-based Complement. Altern. Med. 2013, 2013?
Evid Based Complement Alternat Med. 2013;2013:313142.
doi: 10.1155/2013/313142. Epub 2013 Sep 19.
Grape seed procyanidins in pre- and mild hypertension: a registry study
Gianni Belcaro 1, Andrea Ledda, Shu Hu, Maria Rosa Cesarone, Beatrice Feragalli, Mark Dugall
Author Response
We revised the manuscript with modifications and changes based on the reviewers’ comments.
We send you the revised manuscript together with our point-by-point response.
The changes in the text were highlighted in yellow.
Moreover, the similarity has been revised along the text.
Thanking you in advance for your kind collaboration and suggestions.
Best regards,
The authors
REVIEWER 1
This randomized clinical trial was aimed to evaluate the effects of a 60-day dietary supplementation with Camellia sinensis leaf extract formulated in phospholipids on indicators of adipose tissue dysfunction in overweight or class I obese post-menopausal sedentary women. Twenty-eight women were randomized into two groups: supplemented group (n=14) and placebo group (n=14). The results showed that In regards to the between-group differences over time (β), a statistically significant difference was observed for the Primary endpoint: the respiratory quotient (RQ), the resting energy expenditure (REE), the percentage of carbohydrates oxidation (%CHO), and the percentage of fat oxidation (%LIP) as well as for the Secondary endpoints: waist circumference, insulin, HOMA, and CRP. There are some concerns as listed in the following:
(1) It is better to match the sequence of the statement, e.g. anti/proinflammatory state (CRP, adiponectin) -> anti/proinflammatory state (adiponectin, CRP); liver and kidney function (creatinine, ALT, AST, G-GT)-> liver and kidney function (ALT, AST, G-GT, creatinine) [check all]
Answer: All the sequences of the statements have been rearranged along the whole text.
(2) It is unclear why a significant change of CRP level between the supplemented group and placebo group (p=0.02, Table 4) was not mentioned in the Results and the Abstract.
Answer: The change of CRP level between the supplemented group and placebo group has been mentioned both in the Results and the Abstract.
(3) Keep consistent writing format for the References, including: [check all] e.g.
*Author: R4: Rolland YM, Perry HM 3rd, Patrick P, Banks WA, M.J.; R52: Ryu OH, L.J.L.K.K.H.S.J.K.S.K.N.B.S.C.D.C.K.
*Title: R6: Decreased Plasma Adiponectin Concentrations Are.. vs. Decreased plasma adiponectin concentrations are..
*Journal name: R46: J Am Coll Nutr. vs. J. Am. Coll. Nutr.
*Page number: R4: 1630–6. vs 1630–1636
Answer: All the references have been correct as suggested.
(4) Typos ad others:
*L33-36: no CRP (p=0.02)? Done.
L98: (C reactive protein (CRP), adiponectin) sequence? Done.
L98-99: liver and kidney function (creatinine, alanine ami- 98 notransferase (ALT), aspartate aminotransferase (AST), Gamma-Glutamyl Transferase (G-GT)) Done.
L150: (creatinine, AST, ALT, GGT), Done.
L155: (CRP, adiponectin) Done.
L179: protein. [20]. Done.
L186: (0.94 g/cm3). Done.
L196: oxidation, continuous gas exchange was determined. Done.
L203-204: anti/proinflammatory state (CRP, adiponectin), liver and kidney function (creatinine, ALT, AST, G-GT) Done.
L211-212: we fitted the sentence has been deleted
*L253: and PCR -> and CRP Done.
*L263-267: CRP? (p=0.02) Done.
*L264: MB -> REE (kcal/day) Done.
*L268: Secondary enpoints Done.
*L275: Secondary enpoints Done.
*L415: Clin. Nutr. 2019. [Clin Nutr. 2020 Apr;39(4):1049-1058. doi: 10.1016/j.clnu.2019.05.019. Epub 2019 May 24.]
*L420: U Expert Consultation; 1985; the reference has been checked and is correct
*L422: Nutr. Diabetes 2017, 7 [Nutr Diabetes . 2017 Jan 9;7(1):e238. doi: 10.1038/nutd.2016.38.] the reference has been checked and is correct
*L431: computing; 2017; Done.
*L462: Antioxidants 2021, 10. [Antioxidants (Basel). 2021 Jul 5;10(7):1076. doi: 10.3390/antiox10071076. Done.
*L496: Evidence-based Complement. Altern. Med. 2013, 2013? "2013" in italics has been corrected
Evid Based Complement Alternat Med. 2013;2013:313142.
doi: 10.1155/2013/313142. Epub 2013 Sep 19.
Grape seed procyanidins in pre- and mild hypertension: a registry study
Gianni Belcaro 1, Andrea Ledda, Shu Hu, Maria Rosa Cesarone, Beatrice Feragalli, Mark Dugall
Reviewer 2 Report
The work of Rondanelli et al. it is an interesting nutritional approach to counteract adipose tissue dysfunction in obese and overweight post-menopausal women. The text is well written in all its sections and the results are well expressed, even if the use of some graphs could have made it easier for the reader to interpret the results.
The results (line 236) show that 28 women divided into two groups were enrolled. However, in table 1 there are 29 (Placebo n=14 and GSP n=15). I assume it is a transcription error, please correct where necessary.
Author Response
We revised the manuscript with modifications and changes based on the reviewers’ comments.
We send you the revised manuscript together with our point-by-point response.
The changes in the text were highlighted in yellow.
Moreover, the similarity has been revised along the text.
Thanking you in advance for your kind collaboration and suggestions.
Best regards,
The authors
REVIEWER 2
The work of Rondanelli et al. it is an interesting nutritional approach to counteract adipose tissue dysfunction in obese and overweight post-menopausal women. The text is well written in all its sections and the results are well expressed, even if the use of some graphs could have made it easier for the reader to interpret the results.
The results (line 236) show that 28 women divided into two groups were enrolled. However, in table 1 there are 29 (Placebo n=14 and GSP n=15). I assume it is a transcription error, please correct where necessary.
Answer: The number in Table 4 has been corrected.